Trunk muscle behaviors during the full-cycle stoop and squat lifting tasks

Pan Fumin 1 2
Wang Wei 1 2
Kong Chao 1 2
Lu Shibao 1 2 spinelu@xwhosp.org
1 Department of Orthopedics, Xuanwu Hospital Capital Medical University , Beijing , China
2 National Clinical Research Center for Geriatric Diseases , Beijing , China
Espada Mário
Electronic publication date: 2025 Jan 7
Publication date: 2025
Volume: 13
Electronic Location ID: e18797
Received 2024 Jul 22; Accepted 2024 Dec 11
Copyright: © 2025 Pan et al.
Copyright year: 2025
Copyright holder: Pan et al.
License: This is an open access article distributed under the terms of the Creative Commons Attribution License, which permits unrestricted use, distribution, reproduction and adaptation in any medium and for any purpose provided that it is properly attributed. For attribution, the original author(s), title, publication source (PeerJ) and either DOI or URL of the article must be cited.
License URL: https://creativecommons.org/licenses/by/4.0/

Keywords: Lifting, Stoop, Squat, Trunk muscle, Low back pain

Funding: R&D Program of Beijing Municipal Education Commission KZ202210025038 Chinese Institutes for Medical Research, Beijing CX24PY12 National Natural Science Foundation of China 82102612 This study was supported by the R&D Program of Beijing Municipal Education Commission (Grant No. KZ202210025038), by the Chinese Institutes for Medical Research, Beijing (Grant No. CX24PY12), and by the National Natural Science Foundation of China (Grant No. 82102612). The funders had no role in study design, data collection and analysis, decision to publish, or preparation of the manuscript.

==============================
Background

Lifting is generally considered as a risk factor for low back pain. A thorough investigation of the muscle function during lifting is essential for a better assessment of the potential risk of muscle impairment and towards improvements in lifting strategy. We aimed to compare the activities of the trunk muscles between the stoop and the squat lifting tasks.

Methods

A surface electromyography device was used to measure the muscle activity during the full-cycle squat and the stoop lifting tasks of a 5-kg weight. Each task was divided into four stages: stage 1 was bending forward to reach the weight, stage 2 was lifting the weight up, stage 3 was lowering the weight down, and stage 4 was returning to the upright position. The maximum electromyographic (EMG) activities among different tasks and different stages were compared. Eighteen males aged 20–35 years without low back pain were included, with a mean age of 26.55 ± 2.12 years, body height of 175.18 ± 4.29 cm, body weight of 69.27 ± 4.29 kg, and BMI of 22.56 ± 0.87 kg/m2.

Results

During stoop lifting, the median values of the absolute EMG of the left multifidus were 53.96, 70.32, 51.08 and 64.14 uV from stage 1 to stage 4, which were all non-significantly lower than those during squat lifting for 79.84, 103.64, 71.72 and 95.72, respectively (P > 0.05). The absolute EMG was greatest during stage 2, then during stage 4, stages 1 and 3 came next (Effect size = 0.879, P < 0.001). The median values of the normalized EMG of each muscle during stoop lifting were also non-significantly lower than those during squat lifting at each stage (P > 0.05). The normalized EMG was also greatest during stage 2, then during stage 4, and was lowest during stages 1 and 3 (Effect size = 0.932, P < 0.001).

Conclusion

The trunk muscles were similarly activated during squat and stoop lifting. During lowering the weight down, the trunk muscles were less activated than during extension to the upright position without weight in hands. These results could help to develop subject-specific strategies for lifting tasks to prevent or alleviate occupational low back pain.

Introduction

Low back pain is currently one of the worldwide leading causes for working or life dysfunction, which however lacks success in its treatment due to a lack of knowledge of its complicated mechanism (Cieza et al., 2021; Rezaei et al., 2012). A better understanding of the risk factors for low back pain might hence prevent or alleviate this symptom (Swain et al., 2020). Previously, multiple studies or systematic reviews have identified various risk factors for low back pain, among which weight lifting style is generally considered as an important contributor (Fatoye, Gebrye & Odeyemi, 2019; Heneweer et al., 2011; Parreira et al., 2018). As reported, the incidence of low back pain in industrial workers was as high as 61.6%, which would incur great health and socioeconomic burden (Murtezani et al., 2011).

Previous studies have investigated the factors influencing the lifting kinematics and kinetics, including lifting technique, velocity, direction, weight (Skals et al., 2021; Nolan et al., 2020), and aimed to provide a better understanding of the manner to improve workplace design (Nolan et al., 2020). During lifting, trunk muscles are usually activated to bear external load (Bazrgari, Shirazi-Adl & Arjmand, 2007). As a non-invasive measurement, surface electromyography is usually adopted to measure the muscle activities during lifting (Tsoukos et al., 2021), and could also be implied in the biomechanical model to estimate the low back load (Bazrgari, Shirazi-Adl & Arjmand, 2007). Scholars also demonstrated that subjects with low back pain displayed impairments in trunk muscles compared to those without (Ehsani, Arab & Jaberzadeh, 2017; Gouteron et al., 2022). Accordingly, the muscle functions during lifting should be comprehensively investigated, which is essential for a better assessment of the potential risk of muscle impairment and towards improvements in lifting strategy as well as prevention and treatment for low back pain (Parreira et al., 2018; Tsoukos et al., 2021).

Squat and stoop lifting are the two most popularly investigated lifting strategies, with subjects bending their knees or not to reach the weight regardless of the involvement of the lumbar curvature (Nolan et al., 2020). Generally, squat lifting is safer than stoop lifting in bringing the weight closer to the trunk and, hence, reducing demands on the back muscles to counteract the movements of the external loads (Bazrgari, Shirazi-Adl & Arjmand, 2007). However, previous studies were mainly focusing on the stages during lifting or lowering down the weight (Bieleman et al., 2021; Ghezelbash et al., 2020). No study has investigated the full-cycle lifting task from upright standing to reach the weight (stage 1), lifting the weight up (stage 2), lowering the weight down (stage 3), and then going back to the upright position (stage 4), which could better reflect the lifting task in the daily life. Furthermore, we lift low weight much more frequently than heavy weight in daily life, which is also potentially a risk factor for low back pain but is rarely investigated. Hence, we aimed to compare the activation of multifidus and lumbar erector spinae during the full-cycle squat and stoop low-weight (5 kg) lifting tasks. The following two hypotheses were put forward:

1) During stages 2 and 3 with weight in hands, the trunk muscles would be more activated than during stages 1 and 4 regardless of the movement direction.

2) During stoop lifting, the trunk muscles are more activated than during squat lifting, which is consistent with previous findings (Bazrgari, Shirazi-Adl & Arjmand, 2007; Alemi et al., 2019).

Materials and Methods

Participants

Eighteen adult male participants aged 20–35 years were included, who were young researchers or doctors in the hospital and were included for convenience. The inclusion criteria were that all participants were all asymptomatic from low back or leg pain, and with the Oswestry Disability Index (ODI) of 0. All participants signed consent and the study was approved by the Institutional Review Board (IRB) of Xuanwu Hospital Capital Medical University (No. XW2023-241-002).

Measuring instrument

The wireless FREEEMG® system (BTS Bioengineering, Brooklyn, NY, USA) was used to capture muscle activity during lifting tasks. The bipolar electrodes with a bioelectric signal amplifier allow a total gain of 2,000 and a sampling frequency in the 20–450 Hz range. The skin was prepared following the recommendations of SENIAM (http://seniam.org/), which was firstly shaved and then cleaned with 75% alcohol. The electrodes were placed after the skin was dry. Electromyography data was collected at 1,000 Hz using the BTS MYOLAB® software program. A total of four electromyographic (EMG) electrodes were taped on the left and right multifidus (MF) and longissimus erector spinae (LES). The position of each muscle was also determined following the SENIAM protocol, with 3 and 4 cm from the midline at the levels of L5 and L1 spinal process for MF and LES, respectively. The direction of the electrodes was parallel to the muscles.

Measuring protocol

Two different lifting tasks were performed: stoop and squat lifting. All participants started from the upright standing position. A 5 kg low-weight box was lifted from the ground floor. During stoop lifting, the participants bend their trunk to reach the weight with straight knees (Bazrgari, Shirazi-Adl & Arjmand, 2007; van Dieën, Hoozemans & Toussaint, 1999). The weight was lifted to the abdominal level and then was lowered down to the floor. The participants were then returned to the upright position. During squat lifting, the participants bend their knees to perform the task, with no instruction for the lumbar posture. The two lifting tasks were performed randomly and each participant was instructed to move with their own comfortable velocity. Three trials for each lifting task were performed. Between each two trials, participants rested 30 s to avoid fatigue.

Each task was consequently divided into four stages. Stage 1 was bending forward to reach the weight. Stage 2 was lifting the weight up. Stage 3 was lowering the weight down and stage 4 was returning to the upright position. The four stages were segmented based on observation (Fig. 1). The real-time EMG signals during the four stages were continuously collected.

Figure 1 The four stages during lifting tasks.

Four stages were divided for the stoop (A) and squat (B) lifting tasks. Stage 1 was bending forward to reach the weight. Stage 2 was lifting the weight up. Stage 3 was lowering the weight down and stage 4 was returning to the upright position.

Data processing

The raw data was firstly filtered using the 6th-order high-pass Butterworth filter, with the cut-off frequency at 10 Hz. Then the data was full-wave rectified. The root mean square (RMS) envelopes were computed with a moving window of 100 ms. The maximum value during each stage was recorded and averaged for the three trials for each task. The value during each task and stage was then normalized to the respective maximum value of each muscle to account for the anthropometric difference.

Statistical analysis

Statistical analysis was performed using the IBM SPSS Statistics 19.0 (IBM, Armonk, NY, USA). We firstly verified the normality and variance homogeneity of data by the Kolmogorov-Smirnov test and the Levene’s test, respectively. A three-way repeated analysis of variance (ANOVA) was then conducted to investigate the differences between different tasks, among different muscles and among different stages. The Mauchly’s test was used to test the sphericity assumption. When the sphericity was violated, the Pillai’s trace was used for the averaged tests of significance. Effect sizes (ES) were reported as partial eta squares, with 0.01, 0.06, and 0.14 indicating small, medium, and large effects, respectively (Richardson, 2011). The alpha level was set to 0.05 and corrected using the Bonferroni method for multiple paired comparisons.

Results

The subjects showed a mean age of 26.55 ± 2.12 years, body height of 175.18 ± 4.29 cm, body weight of 69.27 ± 4.29 kg, and body mass index (BMI) of 22.56 ± 0.87 kg/m2.

During stoop lifting, the median values of the absolute EMG of the left multifidus were 53.96, 70.32, 51.08 and 64.14 uV from stage 1 to stage 4, which were all non-significantly lower than those during squat lifting for 79.84, 103.64, 71.72 and 95.72, respectively (Fig. 2). Different muscles showed no significant differences in the absolute EMG activities both during stoop and squat liftings (Effect size = 0.547, P = 0.083; Table 1). For each muscle, the absolute EMG was greatest during stage 2, then during stage 4, stages 1 and 3 came next (Effect size = 0.879, P < 0.001).

Figure 2 The absolute EMG activities of the trunk muscles during lifting.

The absolute EMG activities of the left (A) and right (B) multifidus and left (C) and right (D) erector spinae during different lifting tasks and different stages. The absolute EMG activities during squat lifting were non-significantly greater than during stoop lifting (P > 0.05), and were greatest during stage 2, then during stage 4, and then during stage 1 and 3 (P < 0.05). Rectangle in dash line: statistical significance compared to stage 2, 3 and 4 (P < 0.05). Rectangle in solid line: statistical significance compared to stage 3 and 4 (P < 0.05).

Table 1 Effects of different factors on the muscle activities.

		Absolute	Normalized	
		Effect size	P	Effect size	P	
Main effect	Task	0.286	0.073	0.250	0.098	
	Muscle	0.547	0.083	0.034	0.961	
	Stage	0.879	<0.001	0.932	<0.001	
Interaction effect	Task * Muscle	0.408	0.218	0.158	0.693	
	Task * Stage	0.328	0.339	0.149	0.712	
	Muscle * Stage	0.914	0.333	0.798	0.637	
	Task * Muscle * Stage	0.843	0.536	0.635	0.870	
Note:

P < 0.05 are marked as italic and bold.

The median values of the normalized EMG of the multifidus during stoop lifting were 42.99%, 54.24%, 41.99%, and 47.70% from stage 1 to 4, which were also non-significantly lower than those of 59.29%, 92.47%, 59.01% and 80.13% during squat lifting, respectively (Fig. 3). For each muscle, the normalized EMG was also greatest during stage 2, then during stage 4, and was lowest during stages 1 and 3 (Effect size = 0.932, P < 0.001).

Figure 3 The normalized EMG activities of the trunk muscles during lifting.

The normalized EMG activities of the left (A) and right (B) multifidus and left (C) and right (D) erector spinae during different lifting tasks and different stages. The normalized EMG activities during squat lifting were non-significantly greater than during stoop lifting (P > 0.05), and were greatest during stage 2, then during stage 4, and then during stage 1 and 3 (P < 0.05). Rectangle in dash line: statistical significance compared to stage 2, 3 and 4 (P < 0.05). Rectangle in solid line: statistical significance compared to stage 3 and 4 (P < 0.05).

Discussion

Weight lifting is generally considered as a risk factor for low back pain, which activates the trunk muscles and adds loads on the lumbar spine (Heneweer et al., 2011; Parreira et al., 2018; Nolan et al., 2020). The muscle behaviors during lifting should be comprehensively investigated, which could provide insights into the mechanisms of low back pain. Previous studies demonstrated that squat lifting is safer than stoop lifting, as brings the weight closer to the body trunk (Bazrgari, Shirazi-Adl & Arjmand, 2007; Alemi et al., 2019). Nevertheless, previous studies were mainly focusing on one single stage as lifting the weight up or lowering it down (Nolan et al., 2020; Alemi et al., 2019; Pan et al., 2020). No study has investigated the trunk muscle behaviors during the full-cycle squat and stoop low-weight lifting tasks, which could better reflect the similar task in daily life.

As we investigated the symmetric lifting tasks, the left and right muscles were equally activated. We found that both the absolute and normalized EMG activities of multifidus and erector spinae were greatest during stage 2 with lifting the weight up with a great effect size, which is in line with our cognition, as the subjects have not only to lift the weight up, but also to endure the body weight. However, during stage 4 with extension to the upright position without weight in hands, the trunk muscle activities were also greater than during stage 3 with lowering the weight, which partly rejected our first hypothesis. We assumed that during stage 3, there existed a few needs to enroll the trunk muscles as the inertia of weight also helped to bend the trunk. Hence, during the continuous lifting task, the trunk muscles were dynamically activated and should be separately evaluated, which could provide indications for the establishment of a safe lifting strategy. Nevertheless, we only set the weight to 5 kg, which is far below the lifting capacity of each subject, and the trunk muscles were less required to be activated. The weight of only 5 kg might be far too low to evoke weight-related muscular reactions or adaptations. Further study is therefore needed to increase the lifting weight according to the subject-specific strength capacity.

We also found that the trunk muscle activities during squat lifting were non-significantly greater than during stoop lifting, which rejected our 2nd hypothesis. We assumed that during squat lifting, the trunk was not required to be straight or bended. The subjects would unconsciously to bend back to reach the weight, which would potentially increase the muscle activity. Hence, it could not be concluded from our study that squat lifting is safer than stoop lifting, which might provide insights for deciding the appropriate lifting technique. van der Have, Van Rossom & Jonkers (2019) also demonstrated that squat lifting imposed similar low back loading but higher peak full body musculoskeletal loading compared to stoop lifting. We therefore need to investigate more muscles to provide suggestion for a suitable lifting manner.

Our study owns several limitations. Firstly, we only included 18 male subjects, which could not be representative of the general population. Secondly, we set the lifting weight to 5 kg, which would potentially influence our conclusions. Thirdly, many other factors could influence the lifting behavior, including lifting velocity and weight, which should be investigated in further studies. Fourthly, the measurements performed in the laboratory could not be representative of the tasks performed in the real-world. Fifthly, but not lastly, we did not normalize the position of the weight to the body height, which would potentially change the lifting kinematics and kinetics.

Despite the limitations, we investigated the trunk muscle behaviors during stoop and squat lifting tasks with a low weight of 5 kg. We found that the trunk muscles during stoop lifting were similarly activated compared to those during squat lifting. During lowering the weight, the trunk, muscles were less activated than during extension to the upright position without weight in hands. Our results could help to develop subject-specific strategies for low-weight lifting tasks and for the intervention and prevention of occupational lower back pain.

Supplemental Information

Supplemental Information 1 Raw data.

Additional Information and Declarations

Competing Interests

Author Contributions

Human Ethics

Data Availability

The authors declare that they have no competing interests.

Fumin Pan conceived and designed the experiments, performed the experiments, analyzed the data, prepared figures and/or tables, authored or reviewed drafts of the article, and approved the final draft.

Wei Wang performed the experiments, prepared figures and/or tables, authored or reviewed drafts of the article, and approved the final draft.

Chao Kong analyzed the data, prepared figures and/or tables, and approved the final draft.

Shibao Lu conceived and designed the experiments, authored or reviewed drafts of the article, and approved the final draft.

The following information was supplied relating to ethical approvals (i.e., approving body and any reference numbers):

All participants signed consent and the study was approved by the Institutional Review Board (IRB) of Xuanwu Hospital Capital Medical University (No. XW2023-241-002).

The following information was supplied regarding data availability:

The raw measurements are available in the Supplemental File.

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
