# Peer review of "Trunk muscle behaviors during the full-cycle stoop and squat lifting tasks"

_PeerJ, doi:10.7717/peerj.18797_

## Round 0.1 · original submission · Major Revisions

Dear Authors,

Please revise the manuscript considering the reviewers´ suggestions.

Thank you.

Best regards.

Reviewer 1 ·

Basic reporting

This study is about evaluating muscle demands when performing stoop and squat lifting tasks. The study is well written, yet requires further detail in several sections. Below are my specific comments:

1. Abstract: 'Eighteen males aged 20-35 years without low back pain were included' This can be moved to methods section of the abstract. Please also be more specific about the results here by mentioning the values. E.g., how much greater was the demand during stage 2 than other stages in percentage?

2. Line 33: 'weight lifting' can be changed to 'weight lifting styles'.

3. Line 44: I believe there may be studies that have investigated lifting task cycle from standing upright, lifting/lowering and returning to original state. Please double check the literature.

4. Introduction section needs further depth. Please include a paragraph before the last paragraph introducing the readers about prior studies that have evaluated lifting tasks, common methods used in such evaluations, and their outcomes. Then explain the novelty of this study.

5. Line 49: 'we aimed to compare the activation of multifidus and lumbar erector spinae' Why these specific muscles? Please elaborate. If it is explained in the methods section, you may simply specify the region and the reasoning behind selecting the muscle.

6. Please also include statistics about injury rates in the first paragraph of introduction.

7. Line 51: Since this is a hypothesis, 'were' can be replaced be 'would be'.

8. Line 35: 'Scholars also demonstrated that subjects with low back pain displayed impairments in trunk muscles compared to those without'. Please elaborate further and better relate to the aims of this study. How are impairments in muscles caused by low-back pain related to investigation of demands during lifting?

9. This study includes both trunk flexion-extension and lifting tasks, and the introduction section can introduce both of these tasks, their prevalence in industry, and prior studies that have investigated these tasks.

Experimental design

1. Participants: Please describe the inclusion/exclusion criteria. Why were only male subjects recruited?

2. Line 70: Please explain why these specific muscles were selected by referring to previous studies which may have demonstrated evidence of these muscles getting impacted the most during lifting tasks. Additionally, biceps femoris muscles are also known to be impacted when performing trunk bending and lifting tasks.

3. Line 77: Why was the weight selected as 5 kg? Was this based on literature or pilot studies.

4. How were the four stages segmented, or separated for analysis? Was this based on observation, or any supplemental motion capture system was used to detect specific instances (e.g., start/end of lifting stage)?

5. How was EMG normalized across study participants? Were there any tasks for measuring Maximum Voluntary Contractions (MVCs)? How was this recorded and when? Was a rest period provided?

Validity of the findings

1. The outcomes seem to include absolute values which are directly reported, and without normalization based on individual muscle capacity across participants, I am unsure whether these findings can be meaningful. I would suggest to exclude absolute values and only keep normalized results.

2. This study also includes trunk flexion and returning stages without any load as well as lifting tasks in stages 2 and 3. Please refer to previous studies that have investigated trunk flexion task and compare them with the findings of this study. Below are a few additional references that the authors may consider and could be helpful:

a. van der Have, A., Van Rossom, S., & Jonkers, I. (2019). Squat lifting imposes higher peak joint and muscle loading compared to stoop lifting. Applied Sciences, 9(18), 3794.
b. Kuber, P. M., & Rashedi, E. (2024). Investigating Spatiotemporal Effects of Back-Support Exoskeletons Using Unloaded Cyclic Trunk Flexion–Extension Task Paradigm. Applied Sciences, 14(13), 5564.
c. Wang, Z., Wu, L., Sun, J., He, L., Wang, S., & Yang, L. (2012). Squat, stoop, or semi-squat: A comparative experiment on lifting technique. Journal of Huazhong University of Science and Technology [Medical Sciences], 32, 630-636.

3. Please elaborate on the potential differences in outcomes when performing these tasks in real-world. What differences are expected when the tasks are performed in controlled lab study.

4. Figures 1 and 2: Please mention which of the bars are statistically significant with the help of symbols or letters.

5. Additional figures depicting the task, or photographs can be beneficial to the readers.

Reviewer 2 ·

Basic reporting

Please explain how did you calculate sample size, also because of low sample size, as an alternative to the box plots provided, please prepare a graph presenting individual data as dots connected by lines. Due to the low numbers in your study, more insight into the range of results is required. Here is a sample paper for dots and lines presenting: https://doi.org/10.1111/resp.14810

Experimental design

The sample size is too low.

Validity of the findings

no comment

Reviewer 3 ·

Basic reporting

Figure 2: The text in Figure 2 appears to be cut off, making it difficult for readers to fully understand the content. It would be helpful to revise the figure to improve its readability and ensure that all labels and descriptions are clearly visible.

Experimental design

Low-weight lifting (5kg): It is unclear why a 5kg weight was chosen for this study. Given that low back pain often arises from lifting heavier objects in occupational settings, more justification is needed for focusing on such a low weight. How does this weight relate to typical clinical or workplace scenarios where low back pain prevention is a concern?

Use of Absolute and Median EMG Values: While you provide both absolute and normalized EMG values, it is not clear why both were analyzed in this context. Could you clarify the specific advantages of using absolute EMG values and median values for this study? Further explanation on how each measurement type contributes to the overall interpretation of the results would be helpful.

Sample Size Justification: The study includes 18 participants, but the rationale for choosing this sample size is not clear. It would be helpful to explain how this number was determined.

Validity of the findings

Higher Muscle Activation in Stage 2: In your results, you observed that muscle activation was highest during stage 2 (lifting the weight). However, the discussion does not clearly explain what this increased activation means in terms of biomechanics or clinical significance. Could you provide more insight into why the muscle activity peaks at this stage?

Additional comments

Thank you for your contribution to the understanding of trunk muscle activity during stoop and squat lifting tasks. The topic is highly relevant, particularly regarding its potential implications for preventing low back pain (LBP). However, there are several points regarding the clinical significance and experimental design that I believe need to be clarified or expanded upon.

Reviewer 4 ·

Basic reporting

The manuscript contains conclusions, which are not really supported by the statistical significance calculations of the results, although the also calculated effect sizes provide strong evidence of the conclusions. But the authors do not refer to the effect sizes.
The most problematic point in the manuscript is the absence of abdominal muscles activation data. Further – as already stated by the authors the applied load is very low, which therefore limits the interpretation of the found results.
You provided the dataset in mdx and also tdf- format. These are a special formats. To provide the data in Excel or SPSS format would have been more adequate.

Experimental design

L 57
What was the rationale behind the definition of group size?
L 71-74
Please provide additional figures showing the measurement setup (electrode positions) and also the different lifting tasks.
L 80-81
You say there was not given any instruction to the participants with respect to lumbar posture during squat lifting. If looking at the results squat lifting shows large variations – this might be due to the non-defined lumbar posture and has to be discussed adequately.

L81
Lifting tasks were applied in random order – was this done balanced?

L89-90
Low pass filtering of 10 Hz… Really? Or do you mean high pass filtering?

L92-93
What was the rationale behind the maximum normalization? From my point of view all provided calculations would also been possible with the absolute data. Further, comparisons between tasks are meaningless for the normalized data.

Validity of the findings

Results
L 106-108
You state: Right multifidus, left and right erector spinae showed no significant differences in the absolute EMG activities compared to the left multifidus both during stoop and squat liftings (Effect size=0.547, P=0.083; Table 1).
What do you want to say? Further: in Table 1 no comparisons between sides are provided.

L113-115
Please provide a Table with the respective post hoc tests

Discussion
L119-120
I agree that muscle behavior should be investigated comprehensively, but you should take into account the role of the abdominal muscles as well – and their functional importance for the intra-abdominal pressure.

Table 1
To support the provided result of absence of any side differences, the main effect "Side" should be analyzed also.
Further: as in the ANOVA effect sized larger than 0.14 are considered as large effects, the results ae somehow not really comprehensible. With such large effect sizes, significant differences should be detectable.

Figures
Please be consistent with using diagram headings or not.

Additional comments

Abstract
L24
Conclusion is not supported by the results (at least how they are reported in the abstract)

·

Basic reporting

Language is in objective scientific style.

asic studies on lifting and the methodology of lifting could also better substantiate the approach of the study metodically.: (see)
van Dieën, J. H., Hoozemans, M. J., & Toussaint, H. M. (1999). Stoop or squat: a review of biomechanical studies on lifting technique. Clinical biomechanics, 14(10), 685-696.
Giat, Y., & Pike, N. (1992). Mechanical and electromyographic comparison between the stoop and the squat lifting methods. Journal of safety research, 23(2), 95-105.
data shared structure is ok.
The conclusion is not self-contained with corresponding results to the hypotheses.
This is because no actual hypothesis is formulated, so the conclusion remains without a practically relevant statement and application.

Experimental design

Aims and scope of the journal are basically reached.

The research question is not clearly defined, the relevance is not fully comprehensible. The knowledge gap to be investigated is not clarified, while statements on what additional knowledge can be gathered remain unclear.

The methods are not described in sufficient detail, i.e. how the normalisation of muscle activation was performed. Information on hip angle, upper body length, lever arm etc. is not provided, but is of interest for biomechanical measurements.

Validity of the findings

The data sound valid and robust, though there some methodological short commings mentioned above.

Additional comments

Trunk muscle behaviors during the full-cycle stoop and squat lifting tasks

Basic reporting
- Language is ok.
- The introduction and background fail to provide the relevant context and importance for the conducted cross-sectional study.
- What is the relevance of lifting techniques for back pain? This should be explained.
- Is lifting with light weights relevant?
- Explain the rationale behind studying this particular cohort, why no women were included, and why it did not involve back pain patients. What is special about this decision?
- The rationale for the study is not clearly established.
- The structure is overall good.

Abstract
- Line 14: Please provide a background for your study, not just the aim. Ideally, the background should lead to your aim. Is lifting a particularly dangerous activity concerning back pain? When, why?
- Line 21: Please rewrite the abstract to include the anthropometric data of your participants in the methods section.

Experimental design
- The reason for conducting this study in this manner is not clear. What gap in the existing knowledge on back pain is it supposed to fill? This is not described.
- The results regarding muscle activity during the movement should be adjusted for either the existing lever arm or the upper body length. The authors oversimplify by mentioning methodological weaknesses in the limitations but failing to address them appropriately in the study design.
- Important biomechanical parameters are missing, calling into question the adequacy of the technical standards.

Questions
- As most studies measure only one side of spine muscles, why did you measure both Erector Spinae (ES) muscles in a symmetric loading? What was the expected outcome?
**Intro:**
- Good that you mention low weights, but is this relevant to back pain?
- How do low weights contribute to low back pain?
- How does knowing this help?

Cohort:
- Do you think your population is representative of the typical back pain patient in terms of fitness and movement status?
- Line 78: What does "bend their backs" mean? What was the hip angle? This could refer to the lever arm (upper body length); did you control for this?
- Line 80: Was the distance from the body center controlled for? Considering young and healthy men shouldn't have a problem lifting 5kg at any distance in a squat.
- Line 84: How were the different stages divided? What about the timing and pacing?

---

## Round 0.2 · accepted · Accept

Dear Authors,

Thank you for the work developed in the context of this article.

Please consider the minor suggestions for improvement indicated by reviewer 5 and 2 while in production.

Thank you.

Best regards.

Reviewer 1 ·

Basic reporting

The authors have improved their manuscript based on past comments. I have no further comments to add. I wish the authors all the best in their future endeavors.

Experimental design

NA

Validity of the findings

NA

Additional comments

NA

Reviewer 2 ·

Basic reporting

1. As far as I can see, other reviewers have also requested how you calculated the sample size, but neither mine nor their requests were met.

2. I kindly requested in the first revision that you should providea graph presenting individual data as dots connected by lines. Due to the low paticipants in your study, more insight into the range of results is required. Here is a sample paper for dots and lines presenting: https://doi.org/10.1111/resp.14810. But you did not even answer this request.

3. You stated that you will receive more participants for this study in the future. However you did not provide the sample size calculation. First, you give a sample size. Then, if you cannot reach that sample, you use a term like "pilot study" for this study and then you say "We will receive more participants to reach the sample size we calculated". Even this does not make sense. Why are you publishing the study now if you will receive more participants?

Experimental design

.

Validity of the findings

.

Additional comments

.

Reviewer 3 ·

Basic reporting

no comment

Experimental design

no comment

Validity of the findings

no comment

Additional comments

Thank you for allowing me to review this manuscript. Beyond the three main areas addressed, I don’t have further comments. Overall, the paper is well-constructed, adhering to standards in reporting, experimental setup, and result validity. I believe this study could serve as an important foundational piece of research for understanding and addressing low back pain.

Reviewer 4 ·

Basic reporting

pass

Experimental design

pass

Validity of the findings

pass

·

Basic reporting

ok.

Experimental design

can no longer be changed, I commented on this beforehand.

Validity of the findings

Methodological Detail: Include information on body mechanics, lifting posture specifics, and normalization techniques for muscle activity to ensure replicability.

Still discuss how the findings might apply to broader populations, including women and individuals with back pain.

Clarify Relevance of Results: Temper claims regarding the applicability of findings, given the limited sample size and weight used in lifting tasks.

Additional comments

The manuscript improved substatially. I could get even clearer, if you refine the Introduction a bit more. Better contextualize the study in the literature on lifting biomechanics and low back pain. Clearly articulate the gap and how the study contributes to closing it.